# Analysis of Facial Nerve Functionality and Survival Rates of Patients with Parotid Salivary Gland Carcinoma Submitted to Surgery, Facial Nerve Reconstruction, and Adjuvant Radiotherapy

Wilber Edison Bernaola-Paredes [1,*], Franco Novelli [2], Estefani Albuja-Rivadeneira [2], Adriana Aparecida Flosi [1], Anna Victoria Garbelini Ribeiro [1], Helena Rubini Nogueira [1], Hugo Fontan Köhler [2], Clóvis Antonio Lopes Pinto [3], Kleber Arturo Vallejo-Rosero [4] and Antonio Cassio Assis Pellizzon [1]

1   Department of Radiation Oncology, A.C. Camargo Cancer Center, Sao Paulo 01509-010, Brazil
2   Department of Head and Neck Surgery & Otorhinolaryngology, A.C. Camargo Hospital, Sao Paulo 01509-010, Brazil
3   Department of Anatomic Pathology, A.C. Camargo Cancer Center, Sao Paulo 01509-010, Brazil
4   Department of Oral and Maxillofacial Surgery, Central University of Ecuador, Quito 170129, Ecuador
*   Correspondence: edison.bernaola@gmail.com; Tel.: +55-119-9995-8658

**Abstract:** Background and Objectives: Parotid cancer (PC), when treated surgically, may have associated damage to the functionality of the facial nerve. The role of radiotherapy in the recovery of facial motricity remains controversial. This study aimed to evaluate the impact of radiotherapy (RT) on facial nerve functionality in patients who underwent parotidectomy and facial nerve microsurgical reconstruction. Materials and Methods: Four groups of patients were composed: (a) those who underwent parotidectomy without facial nerve reconstruction and RT; (b) those with nerve reconstruction and without RT; (c) those without nerve reconstruction and RT; and (d) those with nerve reconstruction and RT. Results: 49 patients were male, and 43 were female. A total of 89 underwent parotidectomy, 45 partials, and 44 total. Thus, in nine patients, the sural nerve was used for microsurgical reconstruction. Moreover, 48 patients had a normal facial pattern, 15 with paresthesia, and 29 with permanent paralyses after the House–Brackmann (HB) scale evaluation. Conclusions: The evaluation of nerve functionality after parotidectomy by the House–Brackmann scale is a feasible way to evaluate facial motricity that has already decreased in these patients. Finally, longitudinal studies must be performed to clarify the role of each therapy in the multimodal approach and their clinical impact in facial nerve function.

**Keywords:** facial nerve injury; parotid cancer; radiotherapy

## 1. Introduction

Parotid carcinoma (PC) is a less frequent condition with multifactorial etiology that represents 0.5% of the prevalence and 4–6% of malignant neoplasms that affect the head and neck region (HNC) [1].

The histopathological subtypes described are: mucoepidermoid carcinoma (MC), adenoid cystic carcinoma (ACC), adenocarcinoma (AC), and squamous-cell carcinoma (SCC) as the most prevalent types in these glands, and their treatment is based on surgery in early clinical lesions, in cases with low histological cell differentiation, and according to pathological staging after resection and detection of perineural invasion (PI), in which the compromised nerve should be removed [2,3]. Overall survival (OS) at five years depends on the clinical-pathological staging and has been reported to be almost 70% [2].

Adjuvant RT is indicated when PC has an advanced T-staging (T3, T4), disease-impaired microscopic margins, metastases in the cervical lymph nodes, solid histopathological pattern detected, and the presence of recurrent lesions [3].

The facial nerve (VII cranial pair) is the most important anatomic structure that has a motor function in the muscles of the maxillofacial region, and it could be affected by the tumor; consequently, it must be removed. Thus, with the removal of the facial nerve branches, permanent paralysis becomes imminent. However, there are primary microsurgical reconstruction techniques and nerve grafting that have improved nerve function, thus increasing the quality of life. The type of reconstruction is based on the oncological prognosis, the experience of the surgeon, comorbidities, and the degree of surgical resection, in addition to the reconstruction of the nerve with the remaining branches or nerve grafting. Moreover, of the main sources of grafting traditionally used for reconstruction of the temporal branch of the facial nerve, which is the most compromised, two are outstanding: grafting from the auricular nerve, a branch of the cervical plexus, and that of the sural nerve [3–5].

Permanent facial palsy can occur in up to 7% of patients undergoing parotidectomy [6,7]. Another potential complication is paresthesia, also described in superficial parotidectomy. This has other associated factors, such as tumor extension, duration of surgery, the occurrence of reoperations, intraoperative bleeding, and the need to dissect nerve branches with suspected tumor involvement; this implies manipulation of the nerve branches that would lead to reversible or irreversible paresthesia [6], whereas if there is nerve involvement, the aim is to improve the clinical condition by physical therapy and speech therapy rehabilitation, in addition to other techniques [7].

There is no consensus about the impact of postoperative RT on facial nerve functionality; however, RT is recommended one month after surgery, with a total dose ranging from 54 to 65 Gy (5.5 to 7 weeks for a complete treatment) [8–10]. The aim of the present study is to evaluate the clinical impact of adjuvant RT in patients with the diagnosis of PC, who underwent surgery and facial nerve reconstruction at A.C. Camargo Cancer Center, Sao Paulo, Brazil, in the period between 2008–2019.

## 2. Materials and Methods

### 2.1. Population, Samples, and Ethical Approval

This was an observational, descriptive, and retrospective analysis of electronic medical records of 92 patients diagnosed with parotid carcinoma who underwent multimodal approach to treatment (parotidectomy and RT mainly) during 2008–2019. Four groups were composed of (a) those who underwent parotidectomy without facial nerve reconstruction and RT; (b) those who underwent parotidectomy with facial nerve reconstruction and without RT; (c) those who underwent parotidectomy without facial nerve reconstruction and received RT; and (d) those who underwent parotidectomy with facial nerve reconstruction and received RT. This study was submitted for ethical approval by the Research Ethics Committee (CEP) of the A.C. Camargo Cancer Center, Sao Paulo, Brazil, Protocol No. 4.194.238/2921/20.

### 2.2. Demographic and Clinical Data/House Brackmann Analysis

Demographic data, characteristics of the primary tumor, and the therapy performed were analyzed from the electronic medical records of each patient. As regards facial nerve functionality, the House and Brackmann (HB) grading system was used in the preoperative and postoperative periods in patients with lesions affecting the trunk branches [2,5,6] in six different time intervals: (a) in the preoperative phase; (b) in the immediate postoperative phase; (c) six months of follow up; (d) one year of follow up; (e) two years of follow up; and (f) longer than two years of follow up.

### 2.3. Overall Survival (OS), Disease-Free Survival, and Local Control (LC) Rates

Overall survival (OS), disease-free survival (DFS), and local control (LC) curves were calculated using the Kaplan–Meier, and for differences between them, the log-rank test was performed. Statistical analyses were performed using the SPSS.25 version (SPSS Inc, Sao Paulo, Brazil), with a confidence interval of 95%. Moreover, the Kolmogorov–Smirnov test was used to assess the normality of data collected initially.

### 3. Results

*Demographic, Clinical, and Treatment Data*

A descriptive analysis of demographic and clinical data is shown in Table 1. Of the patients, 53.3% were male, and 46.7% were female. The most common histological subtypes were squamous-cell carcinoma (28.6%), adenoid cystic carcinoma (7.7%), and mucoepidermoid carcinoma (11%). Moreover, 89 patients underwent parotidectomy, among those 49.4% were partial and 50.6% were total. Furthermore, 11.1% of patients underwent microsurgical reconstruction of the facial nerve, and in 90% the sural nerve from the left leg was used.

**Table 1.** Descriptive analysis of demographic, clinical, and treatment data.

| Demographic, Clinical, and Treatment Data | Number of Patients (*n*)/Percentage (%) |
|---|---|
| Sex | Male: 49 (53.3%)<br>Female: 43 (46.7%) |
| Age | ≤60 years: 57 (62%)<br>>60 years: 35 (38%) |
| Smoking | Smoker: 12 (13%)<br>Non-smoker: 40 (43.5%)<br>Ex-smoker: 26 (28.3%)<br>No information: 14 (15.2%) |
| Alcohol consumption | Alcoholic: 6 (6.5%)<br>Non-alcoholic: 45 (48.9%)<br>Social alcohol consumption: 12 (13%)<br>Ex-alcoholic: 7 (7.6%)<br>No information: 22 (23.9%) |
| Histological subtype | Squamous-cell carcinoma (SCC): 26 (28.6%)<br>Adenoid cystic carcinoma (ACC): 7 (7.7%)<br>Mucoepidermoid carcinoma (MC): 10 (11%) |
| Parotidectomy | Partial: 44 (49.4%)<br>Total: 45 (50.6%) |
| Microsurgical reconstruction | Yes: 10 (11.1%)<br>No: 80 (88.9%) |
| Type of nerve flap | Sural nerve: 9 (90%)<br>Other flaps: 1 (10%) |
| Radiotherapy (RT) | Yes: 73 (79.3%)<br>No: 19 (20.7%) |
| RT techniques | IMRT: 44 (60.3%)<br>3D conformal: 23 (31.5%)<br>Others: 6 (8.2%) |
| Total dose (Gy) | ≤60 Gy: 41 (57.7%)<br>>60 Gy: 30 (42.3%) |
| Total sessions (Fx.) | ≤25 fx.: 12 (16.7%)<br>>25 fx.: 60 (83.3%) |
| Chemotherapy (CT) | Yes: 31 (34.1%)<br>No: 60 (65.9%) |
| Patient status after treatment | Death for cancer: 23 (25%)<br>Death for other reasons: 5 (5.4%)<br>Alive with cancer: 4 (4.3%)<br>Alive without cancer: 47 (51.1%)<br>Loss of follow up: 13 (14.1%) |

**Table 1.** *Cont.*

| Demographic, Clinical, and Treatment Data | Number of Patients (*n*)/Percentage (%) |
|---|---|
| Local recurrence (LR) | Local relapse: 24 (26.4%)<br>Non-local relapse: 67 (73.6%) |
| Locoregional recurrence (LRR) | Locoregional relapse: 8 (9%)<br>Non-locoregional relapse: 81 (91%) |
| Metastasis | Distant metastasis: 26 (28.3%)<br>Non-distant metastasis: 65 (70.7%) |
| Local of metastasis | Lung: 20 (21.7%)<br>Bones: 8 (8.7%)<br>Nervous central system: 6 (6.5%)<br>Others: 4 (4.3%) |

Seventy-two patients (78.3%) underwent RT, and the most frequently used technique was intensity modulated radiation therapy (IMRT), with a median total dose of 60Gy (range: 24–71 Gy), delivered in over 25 fractions (median: 30/Range: 10–37 sessions).

Overall, more than 51% of patients are alive without cancer, and 26.4% cases with local relapse and 28.3% with distant metastasis were registered.

After histopathological analysis of surgical specimens of PC in patients submitted to parotidectomy, the pathologic staging was established for initial and advanced disease as a prognostic factor of local recurrence. Most patients showed initial stages (41%) rather than advanced (40.8%) stages. Morphological features such as extracapsular invasion compromised surgical margins, and perineural and angiolymphatic invasion were observed in 8.70%, 15.22%, 22.83%, and 4.4%, respectively. Of the number of lymph nodes compromised by the spread of a primary tumor, more than 3 were detected in 7 patients (30.4%), and a tumor size larger than 4 cm was observed in 21 patients (22.83%), as shown in Table 2.

**Table 2.** Descriptive analysis of morphological criteria of parotid carcinoma in patients submitted to multimodal treatment.

| Morphological Tumor Criteria | Number of Patients | Percentage |
|---|---|---|
| | (*n*) | (%) |
| Pathologic staging (pT) | | |
| Tx | 13 | 14.13% |
| T0 | 2 | 2.17% |
| T1 | 19 | 20.65% |
| T2 | 15 | 16.30% |
| T3 | 10 | 10.87% |
| T4 | 6 | 6.52% |
| T4a | 10 | 10.87% |
| T4b | 8 | 8.70% |
| Unknown | 9 | 9.78% |
| Total | 92 | 100% |
| Neck pathologic staging (pN) | | |
| Nx | 14 | 15.22% |
| N0 | 40 | 43.48% |
| N1 | 12 | 13.04% |
| N2 | 2 | 2.17% |
| N2a | 3 | 3.26% |
| N2b | 5 | 5.43% |
| N3 | 3 | 3.26% |
| N3b | 4 | 4.35% |
| Unknown | 9 | 9.78% |
| Total | 92 | 100% |

**Table 2.** *Cont.*

| Morphological Tumor Criteria | Number of Patients | Percentage |
|---|---|---|
| | (*n*) | (%) |
| Metastasis (pM) | | |
| M0 | 79 | 85.87% |
| M1 | 7 | 7.61% |
| Unknown | 6 | 6.52% |
| Total | 92 | 100% |
| Angiolymphatic invasion | | |
| Yes | 4 | 4.35% |
| No | 86 | 93.48% |
| Unknown | 2 | 2.17% |
| Total | 92 | 100% |
| Vascular invasion (VI) | | |
| Yes | 3 | 3.26% |
| No | 87 | 94.57% |
| Unknown | 2 | 2.17% |
| Total | 92 | 100% |
| Lymph nodes (LNDs) | | |
| Compromised by disease | 23 | 25% |
| Free of disease | 67 | 72.83% |
| Unknown | 2 | 2.17% |
| Total | 92 | 100% |
| Surgical margins | | |
| Compromised by disease | 14 | 15.22% |
| Free of disease | 72 | 78.26% |
| Unknown | 6 | 6.52% |
| Total | 92 | 100% |
| Perineural invasion (PI) | | |
| Yes | 21 | 22.83% |
| No | 69 | 75% |
| Unknown | 2 | 2.17% |
| Total | 92 | 100% |
| Extracapsular extension (ECE) | | |
| Yes | 8 | 8.70% |
| No | 80 | 86.96% |
| Unknown | 4 | 4.35% |
| Total | 92 | 100% |
| Tumor size | | |
| ≤4 cm. | 65 | 70.65% |
| >4 cm. | 21 | 22.83% |
| Unknown | 6 | 6.52% |
| Total | 86 | 100% |
| Number of lymph nodes | | |
| ≤3 | 16 | 69.60% |
| >3 | 7 | 30.40% |
| Total | 23 | 100% |

Table 3 shows the HB scale analysis in patients submitted to parotidectomy and microsurgical facial nerve reconstruction. Analysis was performed in six different phases according to the surgical removal of the initial tumor, which included the period from preoperative to two years of follow up. Heterogeneous results were found based on scale grading, and most of the patients (28 patients) presented Type I over two years after parotidectomy, whereas Type V and VI (facial paralysis) were observed in four patients (8.2%) and one patient (2%), respectively, in the same period.

**Table 3.** Descriptive analysis of House–Brackmann (HB) scale analysis in the different phases after parotidectomy in patients submitted to multimodal treatment.

| HB Grading System | Preoperative Phase N (%) | Postoperative Phase N (%) | Six Months of Follow Up N (%) | One Year N (%) | Two Years N (%) | >2 Years N (%) |
|---|---|---|---|---|---|---|
| I | 68 (87.2%) | 16 (21.1%) | 22 (34.9%) | 27 (48.2%) | 27 (54%) | 28 (57.1%) |
| II | 6 (7.7%) | 20 (26.3%) | 8 (12.7%) | 9 (16.1%) | 7 (14%) | 5 (10.2%) |
| III | 2 (2.6%) | 14 (18.4%) | 11 (17.5%) | 9 (16.1%) | 8 (16%) | 7 (14.3%) |
| IV | 1 (1.3%) | 11 (14.5%) | 10 (15.9%) | 4 (7.1%) | 4 (8%) | 4 (8.2%) |
| V | 1 (1.3%) | 13 (17.1%) | 11 (17.5%) | 6 (10.7%) | 3 (6%) | 4 (8.2%) |
| VI | 0 (0%) | 2 (2.6%) | 1 (1.6%) | 1 (1.8%) | 1 (2%) | 1 (2%) |

After evaluation of nerve functionality by the HB scale in the different phases of oncological treatment, 48 (52.2%) patients had a normal facial pattern, 15 (16.3%) had mild facial paresis, and 29 (31.5%) had facial nerve palsy (permanent paralysis), as shown in Table 4.

**Table 4.** Descriptive analysis of facial pattern (FP) after HB analysis in patients submitted to multimodal treatment.

| Facial Pattern (FP) | Number of Patients ($n$) | Percentage (%) |
|---|---|---|
| Normal | 48 | 52.2% |
| Mild facial paresis | 15 | 16.3% |
| Facial nerve palsy | 29 | 31.5% |
| Total | 92 | 100% |

A statistical association between radiotherapy (RT) and HB scale grading analysis was observed 1 year after the conclusion of treatment ($p = 0.048$). A marginal $p$-value (0.081) was found between HB analysis after two years, and RT was performed, as shown in Table 5.

**Table 5.** Statistical analysis for association between radiotherapy (RT), patient's facial pattern, and HB scale evaluation.

| | | Radiotherapy | | Total | $p$ Value |
|---|---|---|---|---|---|
| | | Yes | No | | |
| | | ($n$) | ($n$) | ($n$) | ($\leq 0.05$) |
| Facial pattern | Normal | 35 (72.9%) | 13 (27.1%) | 48 | 0.28 |
| | Mild facial paresis | 13 (86.7%) | 2 (13.3%) | 15 | |
| | Facial nerve palsy | 25 (86.2%) | 4 (13.8%) | 29 | |
| HB preoperative | Mean $\pm$ SD | 1.27 $\pm$ 0.76 | 1.06 $\pm$ 0.24 | 1.22 $\pm$ 0.68 | 0.28 |
| | Median/Range | 1.00/1–5 * | 1.00/1–2 * | 1.00/1–5 * | |
| HB postoperative | Mean $\pm$ SD | 2.95 $\pm$ 1.54 | 2.65 $\pm$ 1.27 | 2.88 $\pm$ 1.48 | 0.54 |
| | Median/Range | 3.00/1–6 * | 2.00/1–5 * | 3.00/1–6 * | |
| HB 6 months later | Mean $\pm$ SD | 2.88 $\pm$ 1.60 | 2.08 $\pm$ 1.38 | 2.74 $\pm$ 1.58 | 0.12 |
| | Median/Range | 3.00/1–6 * | 1.50/1–5 * | 3.00/1–6 * | |
| HB 1 year later | Mean $\pm$ SD | 2.43 $\pm$ 1.56 | 1.42 $\pm$ 0.67 | 2.21 $\pm$ 1.47 | 0.048 |
| | Median/Range | 2.00/1–6 * | 1.00/1–3 * | 2.00/1–6 * | |
| HB 2 years later | Mean $\pm$ SD | 2.22 $\pm$ 1.46 | 1.22 $\pm$ 0.44 | 2.04 $\pm$ 1.38 | 0.081 |
| | Median/Range | 2.00/1–6 * | 1.00/1–2 * | 1.00/1–6 * | |

**Table 5.** *Cont.*

| | | Radiotherapy | | Total | *p* Value |
|---|---|---|---|---|---|
| | | Yes | No | | |
| | | (*n*) | (*n*) | (*n*) | (≤**0.05**) |
| HB more than 2 years | Mean ± SD Median/Range | 2.21 ± 1.49 1.00/1–6 * | 1.50 ± 1.27 1.00/1–5 * | 2.06 ± 1.46 1.00/1–6 * | 0.16 |

Statistical significance: *p* value ≤ 0.05; SD (standard deviation) * House–Brackmann (HB) scale for nerve functionality evaluation: Type 1, 2, 3, 4, 5, and 6.

Table 6 shows the statistical analysis for the association between RT techniques used in multimodal management, facial pattern, and HB analysis. No statistical significance was found between them in each pattern and in the different phases assessed.

**Table 6.** Statistical analysis for association between RT techniques performed, facial pattern, and HB scale evaluation.

| | | RT Techniques | | | Total | *p* Value |
|---|---|---|---|---|---|---|
| | FP | IMRT (*n*) | 3D (*n*) | Others (*n*) | (*n*) | (≤**0.05**) |
| Facial pattern (FP) | Normal Mild facial paresis Facial nerve palsy | 19 (54.3%) 11 (84.6%) 14 (56%) | 11 (31.4%) 2 (15.4%) 10 (40%) | 5 (14.3%) 0 (0%) 1 (4%) | 35 13 25 | 0.24 |
| HB preoperative | Mean ± SD Median/Range | 1.24 ± 0.63 1.00/1–4 * | 1.39 ± 1.04 1.00/1–5 * | 1.00 ± 0.00 1.00/1 | | 0.68 |
| HB postoperative | Mean ± SD Median/Range | 2.95 ±1.45 3.00/1–6 * | 3.11 ±1.68 3.00/1–6 * | 2.25 ± 1.89 1.50/1–5 | | 0.57 |
| HB 6 months later | Mean ± SD Median/Range | 3.03 ± 1.49 3.00/1–5 * | 2.83 ± 1.76 2.50/1–6 * | 1.00 ± 0.00 1.00/1 | | 0.20 |
| HB 1 year later | Mean ± SD Median/Range | 2.38 ± 1.36 2.00/1–5 * | 2.80 ± 1.88 2.00/1–6 * | 1.00 ± 0.00 2.00/1 | | 0.17 |
| HB 2 years later | Mean ± SD Median/Range | 2.08 ± 1.12 2.00/1–6 * | 2.64 ± 1.95 1.50/1–6 * | 1.00 ± 0.00 1.00/1 | | 0.34 |
| HB more than 2 years | Mean ± SD Median/Range | 1.96 ± 1.12 1.50/1–4 * | 2.85 ± 1.95 3.00/1–6 * | 1.00 ± 0.00 1.00/1 | | 0.22 |

* Statistical significance: *p* value ≤ 0.05; SD (standard deviation); * House–Brackmann (HB) scale for nerve functionality evaluation: Type 1, 2, 3, 4, 5, and 6.

The Spearman test was performed, and a positive correlation between the number of sessions of RT (>25) and the evaluation of the HB scale after 1 year of follow up was established (*ρ*: 0.30/*p* value: 0.05); a positive correlation between the total dose of RT and HB scale analysis at the preoperative phase was statistically significant (*ρ*: 0.28/*p* value: 0.04), as shown in Table 7.

Statistical analysis for correlation between the total dose delivered during RT and the final facial pattern was performed, and doses >60 Gy were related to mild facial paresis after HB scale analysis (*p*: 0.72), whereas for the detection of an association between the number of sessions of RT and FP, in which the highest number of sessions exceeded 25 fractions in all groups, but without statistically significant difference (*p*: 0.64), as shown in Table 8.

**Table 7.** Spearman correlation analysis between the total dose delivered, the number of RT sessions, and HB scale evaluation.

| HB Scale Grading/Radiotherapy Performed (RT) | Total Dose Delivered (Gy) | | | Number of RT Sessions | | |
|---|---|---|---|---|---|---|
| | $\rho$ | *p* Value | Total | $\rho$ | *p* Value | Total |
| HB preoperative | 0.28 | 0.04 | 59 | 0.22 | 0.10 | 60 |
| HB postoperative | 0.09 | 0.50 | 58 | 0.12 | 0.35 | 59 |
| HB 6 months later | 0.13 | 0.37 | 50 | 0.14 | 0.32 | 51 |
| HB 1 year later | 0.29 | 0.06 | 43 | 0.30 | 0.05 | 44 |
| HB 2 years later | 0.24 | 0.13 | 40 | 0.22 | 0.17 | 41 |
| HB more than 2 years later | 0.08 | 0.65 | 38 | 0.03 | 0.86 | 39 |

$\rho$ (rho): Spearman's rank correlation coefficient.

**Table 8.** Statistical analysis for association between the total dose delivered, the number of RT sessions and facial pattern (FP).

| Facial Pattern | Total Dose Delivered (Gy) | | | Number of RT Sessions | | |
|---|---|---|---|---|---|---|
| | Mean | Median | *p* Value | Mean | Median | *p* Value |
| Normal | 57.78 | 60 | | 34.12 | 30 | |
| Mild facial paresis | 60.38 | 60 | 0.72 | 29.92 | 30 | 0.64 |
| Facial nerve palsy | 58.99 | 60 | | 28.60 | 31 | |

OS, DFS, and LC rates were higher in patients submitted to RT for primary tumor management than those who were not, but without statistical significance (*p*: 0.14; 0.44; 0.09, respectively). Figure 1 showed that the OS in patients submitted to partial or total parotidectomy was 50% and 40%, respectively, after 125 months (*p*: 0.89), so more patients who had undergone partial parotidectomy survived than those who had undergone complete removal of the parotid gland.

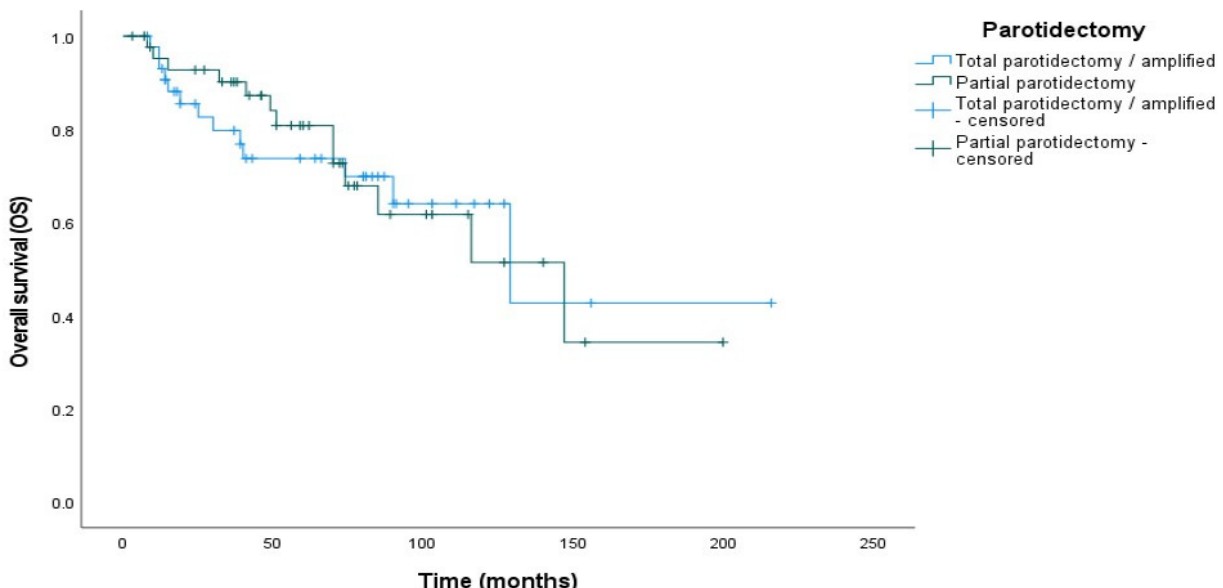

**Figure 1.** Overall survival (OS) and parotidectomy performed in patients submitted to multimodal treatment for parotid carcinoma.

Figure 2 shows the OS of patients who had undergone RT in multimodal management. In those who were submitted to RT, an OS rate of 60% was observed versus 42% in those who did not receive RT (*p*: 0.14). Furthermore, more of those who received RT survived than those who did not receive RT; however, the rate was not statistically significant.

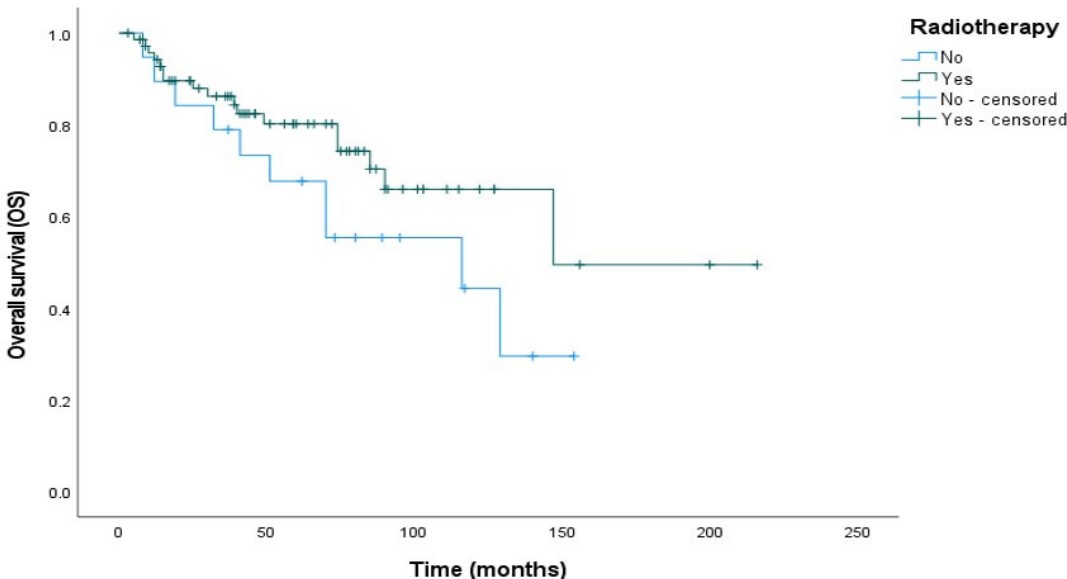

**Figure 2.** Overall survival (OS) and radiotherapy in patients submitted to multimodal treatment for parotid carcinoma.

Figure 3 shows DFS rates considering RT performed, and in patients who received RT, a rate of 42% was observed in comparison with those who did not receive RT (34%) at 17 months after the conclusion of therapy. After performing the log-rank test, no statistical significance was shown (*p*: 0.44).

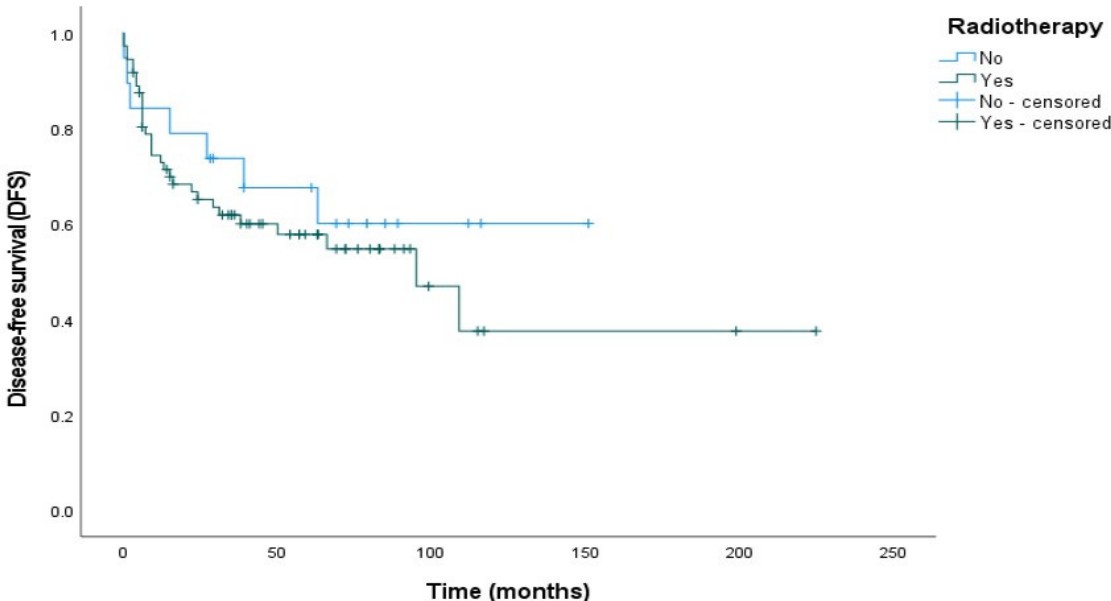

**Figure 3.** Disease-free survival (DFS) and radiotherapy in patients submitted to multimodal treatment for parotid carcinoma.

Figure 4 shows local control (LC) of the disease considering RT performed. In those who did and did not receive RT after 16 months of treatment, the LC rate was observed to be 50% in both groups. However, LC was nine times higher in patients that received RT than in those who did not receive RT, without statistical significance (*p*: 0.09)

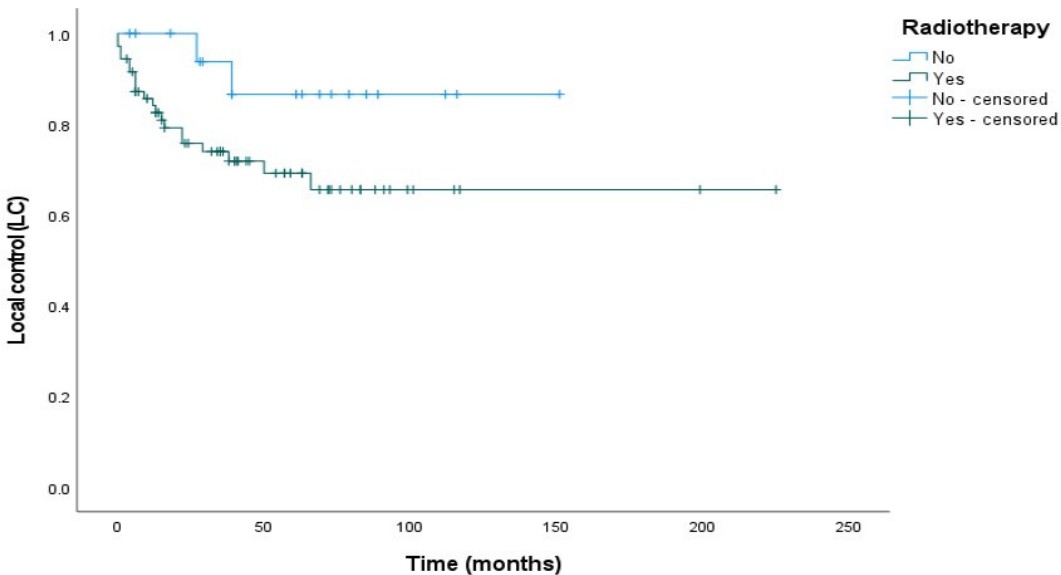

**Figure 4.** Local control (LC) and radiotherapy in patients submitted to multimodal treatment for parotid carcinoma.

## 4. Discussion

Almost 20% of parotid gland tumors are malignant, and those with low-grade or well-differentiated tumors generally exhibit a behavior akin to that of benign lesions diagnosed, such as pleomorphic adenoma (PA), whereas intermediate-grade and high-grade or undifferentiated tumors are more aggressive, they metastasized early to regional lymph nodes and showed a worse prognosis [2]. However, the initial clinical staging of the tumor is not always correlated to the histological type identified after histopathological assessment, so other morphological criteria that have been described as prognostic factors are the degree of cellular differentiation of the primary tumor, size of the tumor, and local extension and/or regional invasion [11]. Our results showed a heterogenous histological subtype with SCC as the type most frequently found in this population (28.6%).

In large tumors (>4 cm in diameter), it may be difficult to identify the facial nerve in its pathological path, so total parotidectomy is recommended for complete removal in these advanced stages to ensure free surgical margins and lymph nodes are compromised [8,12]. The functioning facial nerve should be preserved unless it is infiltrated by the tumor at the time of resection. However, resection of the facial nerve trunk is less common, and when performed, microsurgical reconstruction might be indicated for improvement in the functional and facial pattern. In these cases, the literature has described some options, such as the larger auricular nerve and the sural nerve, as another feasible alternative for reconstructive surgery of the facial nerve [5]. In this study, most of the patients underwent reconstructive surgery with the use of the sural nerve of the left leg as a primary option for facial nerve reconstruction (90%).

For the management of local relapses or distant metastasizing lesions, the multimodal approach has been established to improve local control (LC) and advanced and or recurrent local disease. There is a high rate of transient facial nerve paresis in patients with multiple relapses. Therefore, postoperative RT should be included for better LC and to decrease recurrence rates. Surgery per se showed an improvement in OS rates in five years ranging from 60–80% [13]; however, in cases of locally advanced or recurrent disease, OS and DFS have shown lower rates when surgery is performed alone. Therefore, adjuvant therapies such as RT, chemotherapy (CT), and chemoradiation are still necessary, as well as target therapies that have now shown satisfactory clinical results with higher survival rates. In our results, OS, DFS, and LC rates were higher in patients submitted to RT for primary tumor management than those who were not, but without statistical significance (*p*: 0.29; 0.52; 0.83 respectively).

A recent study [1] evaluated the effect of surgery and postoperative RT in 186 patients diagnosed with PC without differentiation of histological subtypes. OS, LRC, DFS, late toxicity, and quality of life were analyzed, and their results showed improvement in survival rates, clinical condition with side effects reduced, and better quality of life of patients treated; therefore, these were like the results obtained in our study regarding survival, LC rates, and normal facial pattern of patients after HB analysis (Type I, II, and III). Yet, another study, with the aim of evaluating the ability to recover motor functionality after reconstructive surgery of the temporal branch of the facial nerve affected by ACC, used the auricular nerve as primary flap anastomoses with the remaining branches of the injured facial nerve. In addition, the patients were submitted to postoperative RT, and after four years of follow up, patients showed improvement in facial paralysis when compared with values recorded in the immediate postoperative phase [3].

Other studies have described the clinical impact of different therapies, such as CT associated with RT in PC, which was performed when extracapsular spread and compromised surgical margins were shown to avoid the spread of the primary tumor. Thus, the contemporary literature has shown that the concomitant use of CT associated with RT might improve clinical results regarding LC and the spread of the tumor in comparison with only RT performed. However, there is a higher risk for acute toxicity, especially in the elderly. Thus, survival rates in patients who have undergone chemoradiation were higher than rates in those who were only treated with RT [13]. Although in patients with SCC of the parotid gland who were submitted to RT alone when compared with those who had undergone RT and QT associated, no improved LC or LRC rates were found when compared with those of patients who had undergone RT exclusively [14].

The main limitations of this study were the small sample retrieved as a retrospective study and the lack of standardization of data collected for the HB scale in the ninety-two patients included, so it might influence our results or statistical analysis; therefore, it is important to establish, validate, or consider those for clinical practice. Regarding RT performed, patients included were submitted for conventional and IMRT techniques, so the risk of bias is feasible to observe the difference in doses delivered to the target tissues in these techniques is largely described in the contemporary literature. Furthermore, a huge cohort of patients must be included, and a results comparison for each RT technique might be considered in further studies.

## 5. Conclusions

According to our results, the evaluation of nerve functionality in parotid carcinoma by the House–Brackmann scale is a feasible way of evaluating facial motricity that has already decreased in these patients.

Adjuvant radiotherapy showed a positive correlation with the final facial pattern as the clinical outcome measured, which was normal in most of the patients included, and it was associated with the total dose delivered and the number of sessions done. Moreover, survival and local control rates were increased in patients submitted to radiotherapy, irrespective of the pathological staging of the primary tumor in this population.

Finally, longitudinal studies must be performed, which will allow for a better understanding of the role of each therapy in the multimodal approach and regarding nerve dysfunction.

**Author Contributions:** Conceptualization, W.E.B.-P.; data curation, W.E.B.-P., F.N., E.A.-R., A.A.F., A.V.G.R., H.R.N., H.F.K., K.A.V.-R.; formal analysis, W.E.B.-P.; funding acquisition, W.E.B.-P.; K.A.V.-R., investigation, W.E.B.-P.; methodology, W.E.B.-P., H.F.K. and A.C.A.P.; resources, W.E.B.-P., E.A.-R., A.A.F., K.A.V.-R.; validation, W.E.B.-P., F.N., E.A.-R., A.A.F., A.V.G.R., H.R.N., H.F.K., C.A.L.P., K.A.V.-R. and A.C.A.P.; writing—original draft, W.E.B.-P.; writing—review and editing, W.E.B.-P., F.N., E.A.-R., A.A.F., H.F.K., C.A.L.P., K.A.V.-R. and A.C.A.P. All authors have read and agreed to the published version of the manuscript.

**Funding:** This work was supported by the National Council for Scientific and Technological Development (CNPq) of Brazil (140071/2019-9).

**Institutional Review Board Statement:** Regarding ethical approval (8 May 2020), the number protocol was 4.194.238/2921/20 by our Ethics Institutional Committee from A.C. Camargo Cancer Center, Sao Paulo, Brazil.

**Informed Consent Statement:** Informed consent was obtained from all subjects involved in the study.

**Data Availability Statement:** Our data is available for the editors, readers, and any people interested in the process for data collection. If necessary, we will send the link below for full access: https://dados.accamargo.org.br/redcap/redcap_v12.0.30/DataEntry/record_status_dashboard.php?pid=356, accessed on 1 June 2020.

**Acknowledgments:** The authors thank the full Oncological Departments (Radiation Oncology, Head and Neck Surgery and Anatomic Pathology) from A.C. Camargo Cancer Center, Sao Paulo, Brazil, and their teams who supported this work.

**Conflicts of Interest:** The authors declare no conflict of interest.

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
