# Peer review of "Analysis of Facial Nerve Functionality and Survival Rates of Patients with Parotid Salivary Gland Carcinoma Submitted to Surgery, Facial Nerve Reconstruction, and Adjuvant Radiotherapy"

_2038-9582, doi:10.3390/std12020006_

Round 1

Reviewer 1 Report

Authors have studied clinical assessment of facial nerve functionality after various treatment modalities in patients with parotid gland cancer. 

Lines 19-30 Abstract starts well but ends with a conclusion not in concordance with the authors 5. Conclusions. It seems like a copy-paste of the first sentence and does not involve the primary goal of the study to evaluate the RT as a modality.

Table 1 It would be advisable to change the Table 1 arrangement and achieve a clear picture for the readers when browsing through the Left column and finding information in the right column.

Some demographical data is interesting regarding the distribution of the subgroups. In the end, the authors are comparing the subgroup,s so the demographics seem essential compared to survival. One proposal is to use Table 1 for general demographics and afterward wrap up a comparison of the subgroups used for Kaplan-Meier survival.

The mean or median age would be informative. Binarisation seems fine but was not used in any context later in the paper.

Line 104 89 patients (96.7%) - Please change to. Afterward, the authors underwent the division into the type of parotidectomy. Please include the data in Table 1. Therefore several procedures and both types.

Table 2 The table confirms a wide data acquisition pattern but relies on a multitude of data not used for the study itself to support the Discussion or Conclusion. It is more prudent to avoid excessively burdening the reader with a vast amount of data and try to be more precise. The idea is to include only pTNM and not the cTNM.

Moreover, many factors influence the endpoint, especially the survival when data about lymph node cN or margin situation is concerned.

Table 3 The table shows the whole group, preoperatively and during the follow-up. Trends can be observed if the House-Brackmann scale is the defined point of view. It would be more informative if the disease or modality were the illustrated state, which means the HB grading in different groups.

Table 4 The facial pattern report lacks the time frame. As stated, the data is from different phases of oncological treatment and may not represent the best. Table 3 and Table should be synced.

Table 5. Discrete values, p values of less than 0.05, should be easily recognizable. The authors haven't explained why there is a need to present means and medians.

Table 6 Please reformat the table to an appropriate easy-to-read format.

Line 156 Please check whether the appropriate statistical test was included in the Methods section.

Figure 1, Figure 2, Figure 3, Figure 4 Since p values were calculated, it is easier to offer additional info in the context of the explanation below. 

Discussion and Conclusion

I recommend rewriting both sections to more adequately present the study's main message. Patients with RT benefit from the modality, but cannot be quickly confirmed in the present sample. I recommend using a more significant sample rather than a "vast cohort of patients" or inviting readers to think about multicentric studies.

Regarding the facial nerve, the feasibility of HB assessment does not need a particular study. The statistically significant results in the present one do. The only such value is "HB01 year later," as p is 0.048. This might need a better explanation and should receive more focus.

General comment:

Please change values like 01 to 1.

There is also the need for more clarity in the title and the final message in the Discussion and Conclusion. If disease survival and other outcomes are the primary goals, the title should contain them. Should the authors aim to evaluate the FN clinically, that is their focus.

Author Response

Authors have studied clinical assessment of facial nerve functionality after various treatment modalities in patients with parotid gland cancer. 

Lines 19-30 Abstract starts well but ends with a conclusion not in concordance with the authors 5. Conclusions. It seems like a copy-paste of the first sentence and does not involve the primary goal of the study to evaluate the RT as a modality.

Response to reviewer: Done. The full sentence was modified and included a single recommendation to perform longitudinal studies to clarify the role of multimodal treatment, basically the effect of RT.

Table 1 It would be advisable to change the Table 1 arrangement and achieve a clear picture for the readers when browsing through the Left column and finding information in the right column.

Response to reviewer: DONE. The arrangement of Table 1 was modified as requested.

Some demographical data is interesting regarding the distribution of the subgroups. In the end, the authors are comparing the subgroup,s so the demographics seem essential compared to survival. One proposal is to use Table 1 for general demographics and afterward wrap up a comparison of the subgroups used for Kaplan-Meier survival.

The mean or median age would be informative. Binarisation seems fine but was not used in any context later in the paper.

Line 104 89 patients (96.7%) - Please change to. Afterward, the authors underwent the division into the type of parotidectomy. Please include the data in Table 1. Therefore several procedures and both types.

Table 2 The table confirms a wide data acquisition pattern but relies on a multitude of data not used for the study itself to support the Discussion or Conclusion. It is more prudent to avoid excessively burdening the reader with a vast amount of data and try to be more precise. The idea is to include only pTNM and not the cTNM.

Response to reviewer: DONE. Clinical staging and Neck staging as categories from Table 2 were excluded. The pTNM was just considered.

Moreover, many factors influence the endpoint, especially the survival when data about lymph node cN or margin situation is concerned.

Response to reviewer: DONE. These factors were included in Table 2.

Table 3 The table shows the whole group, preoperatively and during the follow-up. Trends can be observed if the House-Brackmann scale is the defined point of view. It would be more informative if the disease or modality were the illustrated state, which means the HB grading in different groups.

Response to the reviewer: This table would like to show the HB scale grading results base don the analysis in the different phase before and after the Parotidectomy performed.

Table 4 The facial pattern report lacks the time frame. As stated, the data is from different phases of oncological treatment and may not represent the best. Table 3 and Table should be synced.

Response to the reviewer: This table just shows a descriptive analysis after the application of facial nerve analysis by HB scale grading for a better understanding of clinical results based on the facial pattern from our population.

Table 5. Discrete values, p values of less than 0.05, should be easily recognizable. The authors haven't explained why there is a need to present means and medians.

Table 6 Please reformat the table to an appropriate easy-to-read format.

Response to reviewer: DONE. The format of this table was modified.

Line 156 Please check whether the appropriate statistical test was included in the Methods section.

Figure 1, Figure 2, Figure 3, Figure 4 Since p values were calculated, it is easier to offer additional info in the context of the explanation below. 

Resposne to reviewer: Agree.

Discussion and Conclusion

I recommend rewriting both sections to more adequately present the study's main message. Patients with RT benefit from the modality, but cannot be quickly confirmed in the present sample. I recommend using a more significant sample rather than a "vast cohort of patients" or inviting readers to think about multicentric studies.

Response to the reviewer: DONE. We recommended in our conclusion the necessity of further longitudinal studies to clarify the role of multimodal treatment, basically of RT performed, in the recovery of facial nerve function and the clinical impact by higher survival rates.

Regarding the facial nerve, the feasibility of HB assessment does not need a particular study. The statistically significant results in the present one do. The only such value is "HB01 year later," as p is 0.048. This might need a better explanation and should receive more focus.

General comment:

Please change values like 01 to 1.

Response to reviewer: DONE.

There is also the need for more clarity in the title and the final message in the Discussion and Conclusion. If disease survival and other outcomes are the primary goals, the title should contain them. Should the authors aim to evaluate the FN clinically, that is their focus.

Response to the reviewer: DONE. The tittle was modified to clarify our main goals : 1) to assess clinically the facial nerve function by HB scale grading after Parotidectomy in patients diagnosed with Parotid Carcinoma and 2) to show the clinical impact (importance) of multimodal treatment, basically in patients undergoing RT by survival outcomes.

Reviewer 2 Report

Thank you for the opportunity to read and critically evaluate the "Clinical Assessment of Facial Nerve..." manuscript. Below are my comments:

Title and abstract

- The title is long and complex. This is not a mistake, but simplifying the title would make it easier to read.

- The abbreviations RT and HB have not been expanded.

Introduction

- The abbreviation HNC should be moved to the end of the sentence.

- The abbreviation ACCHN is unintuitive, it does not follow from the content of the full name.

- The sentence "The histological subtypes..." is too complex and therefore incomprehensible. Please break it down into 2-3 separate sentences.

Materials and Methods

- Combining patients after partial parotidectomy with patients after total parotidectomy into common groups sounds controversial and should be clearly justified. I guess that regardless of whether the deep lobe of the salivary gland was removed, the scope of resection included the facial nerve each time, which should be clearly emphasized.

Results

- Separate characteristics of the individual treatment and control groups (a-d) should be presented in a manner similar to Table 1. This will allow to show similarities and differences that may have influenced the results.

Discussion

- More references to studies by other authors are needed.

Conclusions

- I suggest not to use abbreviations in this section to make it easier for readers who need to quickly get to know the conclusions of the Authors' work.

back matter

- The references section contains too few items, which is a consequence of insufficient support of the introduction and discussion with sources.

In conclusion, I believe that the manuscript needs improvement before publication.

Author Response

Thank you for the opportunity to read and critically evaluate the "Clinical Assessment of Facial Nerve..." manuscript. Below are my comments:

Title and abstract

- The title is long and complex. This is not a mistake, but simplifying the title would make it easier to read.

- The abbreviations RT and HB have not been expanded.

 Response to reviewer: DONE. Corrections suggested were included in the main text.

Introduction

- The abbreviation HNC should be moved to the end of the sentence.

- The abbreviation ACCHN is unintuitive, it does not follow from the content of the full name.

- The sentence "The histological subtypes..." is too complex and therefore incomprehensible. Please break it down into 2-3 separate sentences.

Response to Reviewers: DONE. Full corrections were done as requested. They were highlighted in yellow.

Materials and Methods

- Combining patients after partial parotidectomy with patients after total parotidectomy into common groups sounds controversial and should be clearly justified. I guess that regardless of whether the deep lobe of the salivary gland was removed, the scope of resection included the facial nerve each time, which should be clearly emphasized.

Response to Reviewers: Our main goal of this study is to show the clinical impact of multimodal treatment, basically on RT performed, in patients suffering Parotid Carcinoma who underwent parotidectomy (regardless of the type of Parotidectomy ), and it was assessed by HB grading scale for assessment of facial nerve functionality in  different times before or after surgery and RT and by survival rates as shown in our results.

Results

- Separate characteristics of the individual treatment and control groups (a-d) should be presented in a manner similar to Table 1. This will allow to show similarities and differences that may have influenced the results.

Response to Reviewers: Results were presented according to our main goals to show readers the clinical impact of multimodal treatment in the facial nerve functionality, basically the RT performed, through HB scale grading in different phases after Parotidectomy (simplified in Partial and Total). The table 1 shows demographic and clinical data as general information about these 92 patients.

Discussion

- More references to studies by other authors are needed.

Response to Reviewers: The lack of literature about this issue is one of our main concerns to perform this retrospective study and our results will be encourage further longitudinal studies in our single center and worldwide.

Conclusions

- I suggest not to use abbreviations in this section to make it easier for readers who need to quickly get to know the conclusions of the Authors' work.

Response to Reviewers: DONE. Full abbreviations were removed.

back matter

- The references section contains too few items, which is a consequence of insufficient support of the introduction and discussion with sources.

In conclusion, I believe that the manuscript needs improvement before publication.

Reviewer 3 Report

Explain  what the authors means for paresthesia. Tha facial nerve is a motor nerve and the sensitive nerve of  the region is the great auricular nerve

Author Response

Explain  what the authors means for paresthesia. Tha facial nerve is a motor nerve and the sensitive nerve of  the region is the great auricular nerve

Response to the reviewer: We agree with this definition by the reviewer and the best word to define mild paralysis after Parotidectomy and facial nerve injure among the time is the mild facial paresis and finally the facial nerve palsy (permanent paralysis). Furthermore, mild facial paresis and facial nerve palsy were put instead of paresthesia and facial paralysis as recommended.

Round 2

Reviewer 1 Report

Authors have sufficiently addressed all the comments and proposals. 

Author Response

We appreciate the reviewer’s helpful comments. We have revised our manuscript accordingly, incorporating all the reviewers' comments.
